# Complexity Measures of Heart-Rate Variability in Amyotrophic Lateral Sclerosis with Alternative Pulmonary Capacities

**DOI:** 10.3390/e23020159

**Published:** 2021-01-28

**Authors:** Renata M. M. Pimentel, Celso Ferreira, Vitor Valenti, David M. Garner, Hugo Macedo, Acary S. Bulle Oliveira, Francisco Naildo Cardoso Leitão, Luiz Carlos de Abreu

**Affiliations:** 1Departament of Medicine, Federal University of Santo Paulo (UNIFESP), São Paulo 04024-002, Brazil; doutorcelsoferreira@gmail.com; 2Laboratório de Delineamento de Estudo e Escrita Científica, Centro Universitário FMABC, Santo André 09060-650, Brazil; hugomacedojr@hotmail.com (H.M.J.); nacal@outlook.com (F.N.C.L.); 3Autonomic Nervous System Center (CESNA) Sao Paulo State University, Marília 17525-900, Brazil; vitor.valenti@gmail.com; 4Cardiorespiratory Research Group, Department of Biological and Medical Sciences, Faculty of Health and Life Sciences, Oxford Brookes University, Headington, Oxford OX3 0BP, UK; davidmgarner1@gmail.com; 5Department of Community Health, Faculdade de Medicina do ABC, Santo André 09060-870, Brazil; 6Department of Neuromuscular Diseases, UNIFESP, São Paulo 04022-002, Brazil; acary.bulle@unifesp.br; 7Department of Integrated Health Education, Federal University of Espírito Santo (UFES), Vitória 29040-090, Brazil; luizcarlos@usp.br

**Keywords:** amyotrophic lateral sclerosis, autonomic nervous system, cardiovascular physiology, motor neuron disease, nonlinear dynamics

## Abstract

Objective: the complexity of heart-rate variability (HRV) in amyotrophic lateral sclerosis (ALS) patients with different pulmonary capacities was evaluated. Methods: We set these according to their pulmonary capacity, and specifically forced vital capacity (FVC). We split the groups according to FVC (FVC > 50% (*n* = 29) and FVC < 50% (*n* = 28)). In ALS, the presence of an FVC below 50% is indicative of noninvasive ventilation with two pressure levels and with the absence of other respiratory symptoms. As the number of subjects per group was different, we applied the unbalanced one-way analysis of variance (uANOVA1) test after three tests of normality, and effect size by Cohen’s *d* to assess parameter significance. Results: with regard to chaotic global analysis, CFP4 (*p* < 0.001; *d* = 0.91), CFP5 (*p* = 0.0022; *d* = 0.85), and CFP6 (*p* = 0.0009; *d* = 0.92) were enlarged. All entropies significantly increased. Shannon (*p* = 0.0005; *d* = 0.98), Renyi (*p* = 0.0002; *d* = 1.02), Tsallis (*p* = 0.0004; *d* = 0.99), approximate (*p* = 0.0005; *d* = 0.97), and sample (*p* < 0.0001; *d* = 1.22). Detrended fluctuation analysis (DFA) (*p* = 0.0358) and Higuchi fractal dimension (HFD) (*p* = 0.15) were statistically inconsequential between the two groups. Conclusions: HRV complexity in ALS subjects with different pulmonary capacities increased via chaotic global analysis, especially CFP5 and 3 out of 5 entropies.

## 1. Introduction

Amyotrophic lateral sclerosis (ALS) is a neurodegenerative disease affecting both upper and lower motor neurons [1]. Characterized by progressive neuromuscular atrophy (PMA) with the early involvement of the respiratory system, it may lead to pulmonary collapse that requires mechanical ventilation [2]. Classical ALS corresponds to 70% of cases, and can be classified as bulbar (33%) or spinal (66%) depending on symptom onset [1].

The weakness and fatigue of respiratory muscles eventually induce respiratory insufficiency, which is the main cause of death in this disease [3]. The second most frequent cause is related to cardiovascular diseases [4]. There is recent evidence that ALS diagnosis can be established before symptoms become apparent, such as through terminal phrenic nerve latency [3,5] that correlates with respiratory symptoms and through forced vital capacity (FVC) [3].

Patients with motor neuron disease (MND) have a higher risk of sudden death, and the heart is often affected in these patients. There is evidence that autonomic involvement can result in sympathetic hyperactivity, increasing the risk of sudden death; however, there is a lack of data on this cause [6].

Heart-rate variability (HRV) can be assessed from RR-intervals. These are derived from an electrocardiographic PQRST-signature and can deviate in chaotic or irregular manner. HRV is an inexpensive, noninvasive, and reliable method of monitoring sympathetic and parasympathetic balance [7,8]. Variability analysis represents a novel means to evaluate and treat individual to provide diagnostic, prognostic and pathophysiological information [9]. 

HRV verification checks are vital to detect disorders of cardiac autonomic dysfunction in patients with ALS to avoid sudden death or other conditions that lead to a decrease in life expectancy [10]. Despite evidence of autonomic dysfunction in these patients, few managed to explain result divergence, sometimes showing sympathetic hyperactivity and sometimes demonstrating parasympathetic hypoactivity. Merico and Cavanatato [11], and Pinto [12] related this fact to phenotypic heterogeneity, highlighting bulbar involvement, while Pinto [13] and Baltadzhieva [14] related it to the stage of the disease, such as the need for ventilators.

In this study, we evaluate the complexity of HRV in ALS patients [15,16]. We diagnosed these according to pulmonary capacity, specifically FVC. Pimentel et al. presented no significant correlation between HRV and pulmonary capacity by linear methods, and indicated a significant decrease in the SD2 index in the subjects who needed ventilatory support [17]. The absence of significant differences in time (mean SD of normal RR intervals, percentage of adjacent RR intervals with a difference of duration of 50 ms, and the root of square successive differences between adjacent normal RR intervals) and frequency (high frequency, low frequency, and low frequency to high frequency ratio) domains was because nonlinear indices were highly sensitive at detecting changes. In addition, nonlinear behavior was predominant in human physiological systems owing to their complex and dynamic nature [16].

We assumed that the subjects’ RR intervals performed in a nonlinear way akin to diabetes mellitus [18], chronic obstructive pulmonary disease (COPD) [19], and epilepsy [20,21] as dynamical conditions which can be quantified and assessed by applying globally chaotic algorithms [22] and other nonlinear techniques. By assessing the level of chaotic response via these techniques derived from nonlinear dynamical analysis, we can assess the probability of attaining “dynamical diseases” [23].

We can assume that impaired respiratory function is important, but this article highlights the importance of HRV as a prognostic predictor for ALS in order to detect early abnormalities, estimate prognosis, and define treatment strategies; therefore, the objective of this study was to evaluate the complexity of HRV in ALS patients with different pulmonary capacities.

## 2. Materials and Methods

### 2.1. Study Population

In this transversal study, we examined 57 patients with a medical diagnosis of ALS spinal. For the present study, recruitment was performed in the Brazilian Association of Amyotrophic Lateral Sclerosis (ABRELA) associated with the Department of Neuromuscular Diseases of UNIFESP from March to December 2016. This group was divided and classified according to pulmonary capacity. ALS was confirmed by the identification of features present in the El Escorial criteria for the diagnosis of ALS [24], supported by de Carvalho et al. [25], who signed (patient or caregiver) authorization to participate in the study. Patients with ALS bulbar and congenital abnormalities, heart diseases, pulmonary malformations, and patients who used medications interfering with the autonomic nervous system response, such as antiarrhythmic medicines and insulin, were excluded from the study.

### 2.2. Pulmonary Evaluation

Spirometry examination was performed to verify the vital capacity. Patients were classified according to their pulmonary capacity (FVC) into two groups: (1) FVC > 50% (*n* = 29) and (2) FVC < 50% (*n* = 28). Through these parameters, patients with less than 50% FVC presented the need for non-invasive ventilation. (Table 1). In neuromuscular-disease conditions such as ALS, the presence of FVC below 50% of the predicted was indicative of non-invasive ventilation with two pressure levels, even in the absence of respiratory symptoms [26]. Patients with bulbar ALS were excluded from the study because it is common that bulbar involvement can cause facial weakness, which further impaired the accurate assessment of vital capacity.

### 2.3. Experimental Protocol and HRV Analysis

Data collection was commenced at room temperature between 21 and 25 °C, and with humidity between 50 and 60%. Patients were instructed to not ingest alcohol, caffeine, or other autonomic stimulants for 24 h prior to evaluation. Data collection was achieved individually between 18:00 and 21:00 to standardize circadian influences [27]. Patients were instructed to remain in the sitting position, at rest, and to avoid conversation during the experiment. Individuals with FVC < 50% were not under ventilation support during recording.

After the initial evaluation, the heart-monitor belt was then placed over the thorax, and aligned with the distal third of the sternum and the Polar RS800CX heart-rate receiver (Polar Electro, Finland). RR intervals were recorded with a sampling rate of 1 kHz. They were then transferred to Polar Precision Performance software (v. 3.0, Polar Electro, Finland). This software allowed for heart-rate visualization and the extraction of a file relating to a cardiac period (RR interval) in a txt file. After digital filtering supplemented with manual filtering to eliminate artefacts, 1000 RR intervals were applied for data analysis. We included only series with >95% sinus beats [28].

### 2.4. Nonlinear Analysis

Nonlinear HRV analysis included detrended fluctuation analysis (DFA) [29], chaotic global assessments [30,31], high spectral entropy [19], high spectral DFA [32], spectral multitaper method (sMTM) [22], chaotic forward parameters (CFP1 to CFP7), entropic assessments; Shannon entropy [33], Renyi entropy [34,35], Tsallis entropy [36,37,38], approximate entropy (ApEn), sample entropy (SampEn) [39,40], and Higuchi fractal dimension (HFD). Details about the abovementioned methods were previously reported [19,22,29,30,31,32,33,34,35,36,37,38,39,40].

### 2.5. Effect Size

To quantify the magnitude of difference between protocols for significant differences, effect size was calculated using Cohen’s *d* for significant differences (*p* < 0.005) [8]. Large effect size was considered for values > 0.9, medium for values between 0.9 and 0.5, and small for values between 0.5 and 0.25 [41,42].

### 2.6. Principal-Component Analysis (PCA)

Principal-component analysis (PCA) [43] is a multivariate technique for analyzing the complexity of high-dimensional datasets. PCA is useful when (a) sources of data variability need to be explained, and (b) it reduces data complexity and assesses data with fewer dimensions through this. Primary PCA goals are to rationalize the sources of data variability and to represent data with fewer variables while sustaining the majority of the total variance.

## 3. Results

### 3.1. Mean, Standard Deviations, Unbalanced One-Way Analysis of Variance (uANOVA1), and Cohen’s D

Mean and standard deviations were calculated for chaotic globals, DFA, Higuchi fractal dimensions, and the five entropies. After assessments for normality (Ryan–Joiner, Anderson–Darling, and Lilliefors), these were tested for significance by uANOVA1 [44]. The number of subjects in the two assessed groups was unequal; therefore, standard one-way analysis of variance (ANOVA1) [45] or Kruskal–Wallis [46] techniques were unsuitable.

First, regarding chaotic globals (Figure 1 and Table 2), CFP4–CFP7 revealed the most significant changes in response to the two conditions (FVC > 50% and FVC < 50%). They all gave a significant increase in parametric response with the exception of CFP7. Effect sizes corresponded with uANOVA1 *p* values, since four previously cited chaotic globals had low *p* values (*p* ≈ 0.002) and high Cohen’s *d* (*d* ≈ 0.9). The other chaotic globals (CFP1, CFP2, and CFP3) had high *p* values and low Cohen’s *d* effect sizes, and so were not considered for statistical significance.

Next, DFA was significant with *p* value (*p* = 00358) and Cohen’s *d* (*d* = 0.57). Moderate decrease was observed, which was expected. DFA responded in the opposite manner to chaotic globals, fractal dimensions, and entropies.

Then, the five entropies (approximate, sample, Shannon, Renyi, and Tsallis) that were calculated provided a significant increase with *p* values (*p* ≈ 0.0002) and Cohen’s *d* (*d* ≈ 1.0) (Table 3). Sample entropy had a particularly high Cohen’s *d* value (*d* = 1.22) and largely compensated for prevalent bias in approximate entropy.

Lastly, the Higuchi fractal dimension was not significant for all values of *K_max_*. The highest significance was achieved for a *K_max_* of 40, where *p* ≈ 0.15 or 15% (Figure 2 and Figure 3). This was inconsequential because we set the level of significance to *p* < 0.005 (or 0.5%) throughout [32,37].

### 3.2. Multivariate Analysis via PCA

CFP4 had the first (PC1) (0.486) and second (PC2) (0.061) principal components. Next, CFP5 had PC1 (0.458) and PC2 (0.207). However, CFP6 had PC1 (0.486, identical to CFP4 PC1) and PC2 (−0.035 less than CFP4 PC2). Lastly, CFP7 had PC1 (−0.416) and PC2 (−0.314).

Only the first two components were considered due to the steep scree plot. Cumulative influence as a percentage was 59.9% for PC1, and 99.4% for the cumulative total of PC1 and PC2. PC2 had 39.5% influence. So, CFP4, which applied two of the three chaotic global techniques, was the optimal and most robust overall combination regarding influencing the correct outcome. The two applied techniques were *s*MTM and *hs*DFA. Difference, however, was marginal (Table 4).

Regarding PCA values for the five entropies, a clear pattern was revealed. Shannon, Renyi, and Tsallis entropies performed similarly. ApEn and SampEn were analogous (Table 5). The first three components needed to be considered due to the relatively lesser steep scree plot compared to chaotic globals. Cumulative influence as a percentage was 65.4% for PC1, and 89.3% for the cumulative total of PC1 and PC2. PC2 had 23.8% influence. Cumulative influence as a percentage was 99.3% for the total of PC1, PC2, and PC3, and PC3 alone had 10.0% influence. Shannon entropy had PC1 (0.474), almost the same as the PC1 for Renyi (0.476) and Tsallis (0.475); PC2 (0.246) was again similar to PC2 for Renyi (0.223) and Tsallis (0.243). Lasly, PC3 (−0.219 was similar to PC3 for Renyi (−0.230) and Tsallis (−0.220).

Next, ApEn and SampEn performed in the same way. Once more, the first three components needed to be considered. ApEn had PC1 (0.301) whereas SampEn was 0.351, and PC2 (−0.662) was again similar to SampEn at −0.591. Lastly, PC3 was 0.052, whereas SampEn was −0.004. Since PC1 had the greatest influence, the SampEn algorithm overall slightly outperformed ApEn.

With regards to DFA (Table 5), we calculated PCA to assess its influence despite being insignificant on the uANOVA1 test, and it peformed moderately on PCA.

## 4. Discussion

### 4.1. Statistical Results

CFP4 and CFP6 performed best on chaotic global analysis. CFP5 and CFP7 performed adequately, but CFP7 was different in that the transition from FVC > 50% to FVC < 50% encouraged a significant decrease. Low *p* values corresponded with high effect sizes, as was expected.All five entropies performed well and attained an increase in chaotic response from FVC > 50% to FVC < 50% cohorts. Low *p* values matched with high Cohen’s *d* effect sizes, with sample entropy having a particularly high effect size (1.22) and was, therefore, of high significance.DFA did not attain a similarly significant response as those in the chaotic globals or entropies. It failed to achieve significance on both uANOVA1 and the Cohen’s *d* effect sizes. It did, nevertheless, attain a decreased chaotic response, which is to be expected since the parameter usually responds in the contradictory manner.

### 4.2. Principal-Component Analysis

Those that were significant were CFP4, CFP5, CFP6 and CFP7. They all performed similarly with regards to PC1. CFP7 presented the opposite response to that of the others, so we did not discuss this combination further. PC2, CFP4, and CFP6 responded similarly, but CFP5 outperformed both on the second component; therefore, CFP5 was the statistically best and most influential performing chaotic global.Regarding five entropies and multivariate analysis through PCA, Shannon, Renyi, and Tsallis entropies performed similarly on PC1, PC2, and PC3. We only considered the first three components due to a moderately steep scree plot. Approximate and sample entropies also performed similarly on PC1, PC2, and PC3, with sample entropy slightly outperforming approximate entropy due to higher responses on the PC1 (0.351), the most influential component.DFA was not significantly altered with regards to mean and standard deviations (see above). It was, nevertheless, moderately influential on the basis of PCA, and was not considered further.

### 4.3. Signal Processing

Different fractal-dimension algorithms could be applied to data in addition to the Higuchi fractal dimension, for example, those by Katz [47] and Castiglioni [48].The weighting of chaotic globals could be adjusted when calculating seven permutations of chaotic forward parameters.Parameters of the multitaper-method (MTM) spectral technique enforced by the chaotic globals could be modified: (a) sampling frequency, (b) time bandwidth for DPSS, (c) FFT length, (d) Thomson’s nonlinear combination method.The length of the datasets could be increased beyond 1000 RR, and the number of subjects in the study could be increased.

### 4.4. Clinical Interpretation

Nonlinear HRV analysis can significantly improve HRV utilization for risk stratification [38].Reduced HRV complexity is frequently achieved during stressful situations and disorders [34]. We expected a diminished HRV chaotic response in ALS patients with FVC < 50%; however, we revealed the opposite.Previous evidence indicated that the elevated values of nonlinear HRV were not always associated with physiological wellbeing [49,50,51]. Our results demonstrated increased complexity of HRV in a pathological state of ALS. This finding helps to break the paradigm that increased HRV complexity is related to better physiological health.Our results promote the hypothesis that ALS patients with FVC < 50% presented increased HRV complexity because of their determination to overcome impaired healthy condition. We encourage further studies to expand this technique by using anti-inflammatory biomarkers.

Transition from FVC > 50% to FVC < 50% encourages significant increase rather than decrease in chaotic response. The most apt parameters to be used when assessing this chaotic status are CFP5 (function of *hs*DFA alone) from chaotic global permutations, and Shannon, Renyi, or Tsallis entropy. We discourage the use of approximate and sample entropies despite their significance, especially with regards to Cohen’s *d* for sample entropy. These two tests are very parameter-dependent (M, embedding dimension; r, tolerance) and, therefore, difficult to set to attain appropriate and optimal assessment. It is also normal to apply (although different data respond in different ways) and M = 2 and r = 0.2 where 0.2 represents 20% of the standard deviation of RR interval time-series data. These standard values were applied here, as is usually the case [52].

Consequently, the increase in chaotic response during the pathological state should unexpectedly reduce the risk of developing dynamical diseases in subjects with ALS and FVC < 50%.

## 5. Conclusions

HRV complexity in ALS subjects with different pulmonary capacities increased through chaotic global analysis, especially CFP5 and 3 of 5 entropies. This study suggests that nonlinear indices are highly sensitive in detecting changes in human physiological systems; therefore, further studies should be carried out in order to use HRV verification in detecting cardiac-autonomic-dysfunction disorders in ALS patients and early abnormalities, estimating prognosis, and defining treatment strategies.

## Figures and Tables

**Figure 1 entropy-23-00159-f001:**
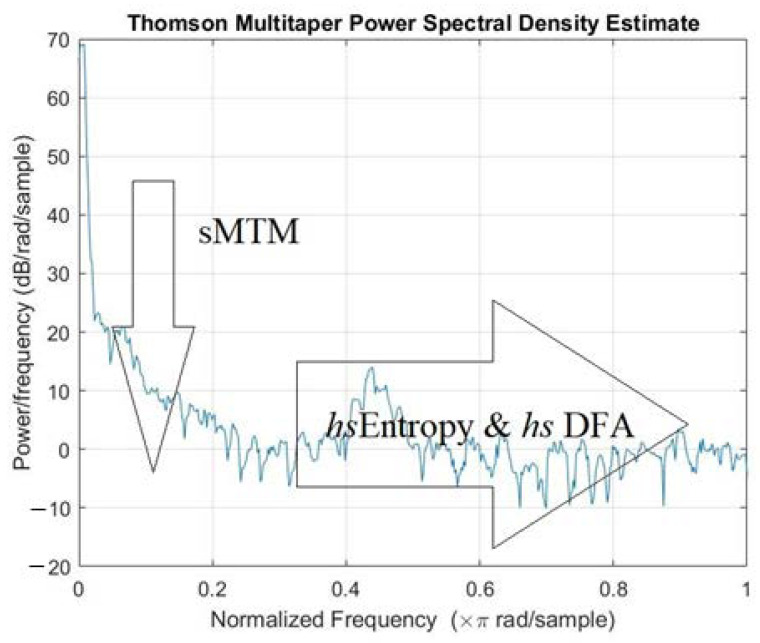
The multi taper method (MTM) power spectrum of a time series of 1000 RR intervals in experimental subject (FVC < 50%). sMTM was the area beneath the spectrum but above the baseline formed by broadband noise as signal becomes chaotic. High spectral entropy (*hs*Entropy) and high spectral detrended fluctuation analysis (*hs*DFA) were derived by applying Shannon entropy and DFA functions to MTM power spectrum. Parameters for MTM power spectra were set at (i) sampling frequency, 1 Hz; (ii) time bandwidth for Discrete Prolate Spheroidal Sequences (DPSS), 3; (iii) Fast Fourier Transform (FFT) length, 256; (iv) Thomson’s “adaptive” nonlinear combination method to combine individual spectral estimates.

**Figure 2 entropy-23-00159-f002:**
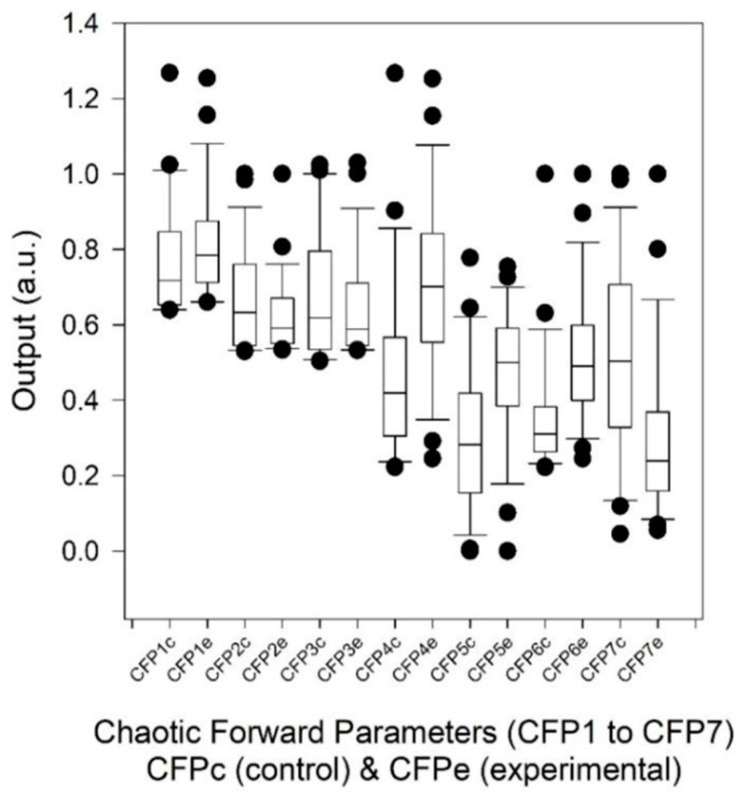
Values of chaotic forward parameters (CFP1 to CFP7) for control (FVC > 50%) and experimental (FVC < 50%) subjects with 1000 RR intervals. Point closest to zero, minimum; point farthest away, maximum. Point next closest to zero, 5th percentile; point next farthest away, 95th percentile. Boundary of box closest to zero, 25th percentile; line within box, median; boundary of box farthest from zero, 75th percentile. Difference between these points, interquartile range (IQR). Whiskers (or error bars) above and below box, 90th and 10th percentiles, respectively.

**Figure 3 entropy-23-00159-f003:**
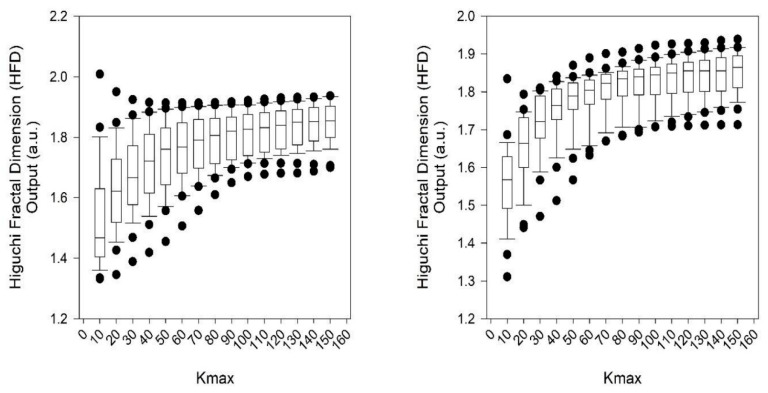
Box-and-whiskers plot for Higuchi fractal dimension of RR intervals of control (**left**: *n* = 29 with FVC > 50%) and experimental (**right**: *n* = 28 with FVC < 50%) subjects calculated multiple times from 10 to 150 in equidistant units for different levels of *Kmax*. Saturation point achieved at *Kmax* 40 (Figure 4). Point closest to zero, minimum; point farthest away, maximum. Point next closest to zero, 5th percentile; and point next farthest away, 95th percentile. Boundary of box closest to zero, 25th percentile; line within box, median; boundary of box farthest from zero, 75th percentile. Difference between these points, interquartile range (IQR). Whiskers (or error bars) above and below box, 90th and 10th percentiles, respectively.

**Figure 4 entropy-23-00159-f004:**
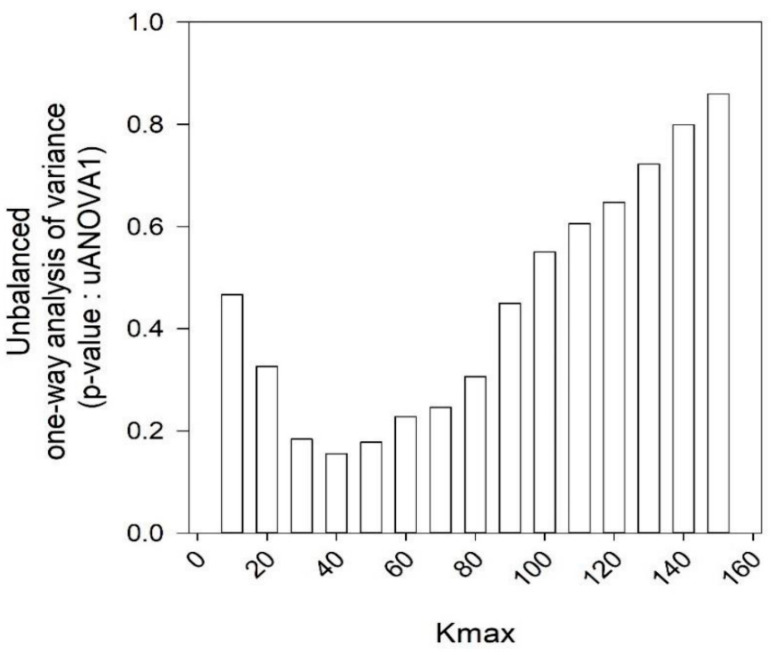
uANOVA1 for Higuchi fractal dimension at varying levels of *Kmax* between 10 and 150 at equidistant intervals of 10. Data control set had FVC > 50% (*n* = 29) and experimental dataset had FVC < 50% (*n* = 28). The lowest *p* value and hence the greatest significance was accomplished for *Kmax* 40 (*p* ≈ 0.15, ≈ 15%).

**Table 1 entropy-23-00159-t001:** Characterization of sample subjects. FVC, forced vital capacity.

Variable	FVC > 50%	FVC < 50%
*n*	29	28
Age (years)	57.8 ± 10	54.7 ± 9
Height (m)	1.65 ± 0.07	1.66 ± 0.07
Mass (kg)	67.3 ± 13.9	63.1 ± 9.6

FVC: forced vital capacity.

**Table 2 entropy-23-00159-t002:** Mean values, standard deviation, unbalanced standard one-way analysis of variance (ANOVA1), and Cohen’s *d* effect size for CFP1 to CFP7 for control (FVC > 50%) and experimental (FVC < 50%) subjects.

Chaotic Global	Mean ± SD FVC > 50% (*n* = 29)	Mean ± SD FVC < 50% (*n* = 28)	uANOVA1 (*p* Value)	Effect Size Cohen’s *d*
CFP1	0.7718 ± 0.1495	0.8269 ± 0.1569	0.1797	0.35
CFP2	0.6633 ± 0.1372	0.6231 ± 0.1046	0.2196	0.33
CFP3	0.6736 ± 0.1659	0.6530 ± 0.1457	0.6211	0.13
CFP4	0.4889 ± 0.2429	0.7144 ± 0.2476	<0.001	0.91
CFP5	0.3127 ± 0.2014	0.4774 ± 0.1844	0.0022	0.85
CFP6	0.3651 ± 0.1632	0.5258 ± 0.1831	0.0009	0.92
CFP7	0.5033 ± 0.2651	0.2976 ± 0.2266	0.0027	0.83

FVC: Forced Vital Capacity; CFP 1 to CFP7: Chaotic Forward Parameters.

**Table 3 entropy-23-00159-t003:** Mean values, standard deviation, unbalanced ANOVA1, and Cohen’s *d* effect size for five entropic and DFA measures for control (FVC > 50%) and experiemental (FVC < 50%) subjects.

Entropy or DFA	Mean ± SD FVC > 50% (*n* = 29)	Mean ± SD FVC < 50% (*n* = 28)	uANOVA1 (*p* Value)	Effect Size Cohen’s *d*
Approximate	0.6435 ± 0.1829	0.8033 ± 0.1417	0.0005	0.97
Sample	0.6055 ± 0.1971	0.8176 ± 0.1445	<0.0001	1.22
DFA	0.4735 ± 0.2538	0.3279 ± 0.2569	0.0358	0.57
Shannon	0.5525 ± 0.1270	0.6809 ± 0.1348	0.0005	0.98
Renyi	0.9846 ± 0.0051	0.9898 ± 0.0050	0.0002	1.02
Tsallis	0.5930 ± 0.1173	0.7131 ± 0.1231	0.0004	0.99

**Table 4 entropy-23-00159-t004:** Principal-component analysis for CFP for 7 groups of 28 experimental subjects with FVC < 50%.

Chaotic Global	PC1	PC2	PC3	PC4	PC5	PC6	PC7
CFP1	0.354	−0.414	0.122	−0.556	0.461	0.146	−0.381
CFP2	0.019	−0.595	0.663	0.262	−0.096	−0.079	0.349
CFP3	0.139	−0.573	−0.556	0.407	−0.197	−0.177	−0.328
CFP4	0.486	0.061	0.003	−0.442	−0.602	−0.428	0.140
CFP5	0.458	0.207	0.255	0.305	−0.337	0.598	−0.343
CFP6	0.486	−0.035	−0.365	0.090	0.271	0.292	0.680
CFP7	−0.416	−0.314	−0.194	−0.400	−0.436	0.561	0.163

FVC: Forced Vital Capacity; CFP 1 to CFP7: Chaotic Forward; Parameters PC1 to PC7: Principal Component.

**Table 5 entropy-23-00159-t005:** Relevant principal-component analysis for five entropies and detrended fluctuation analysis (DFA) of 28 experimental subjects with FVC < 50%.

Entropy (or DFA)	PC1	PC2	PC3	PC4	PC5	PC6
Approximate	0.301	−0.662	0.052	−0.651	0.211	< 0.001
Sample	0.351	−0.591	−0.004	0.698	−0.203	< 0.001
DFA	0.330	0.208	0.921	0.011	−0.015	< 0.001
Shannon	0.474	0.246	−0.219	0.104	0.467	−0.662
Renyi	0.476	0.223	−0.230	−0.274	−0.767	−0.084
Tsallis	0.475	0.243	−0.220	0.061	0.329	0.745

FVC: Forced Vital Capacity; Parameters PC1 to PC7: Principal Component; DFA: Detrended Fluctuation Analysis.

## Data Availability

The used and analyzed datasets during the present study are available from the corresponding author on reasonable request.

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
