# Peer review of "Complexity Measures of Heart-Rate Variability in Amyotrophic Lateral Sclerosis with Alternative Pulmonary Capacities"

_entropy, 2021, doi:10.3390/e23020159_

Round 1

Reviewer 1 Report

  1. There may be a problem with the reliability of the FVC measurement. If the bulbar palsy is severe in ALS patients, the reliability of the FVC value is problematic. For this purpose, the patient characteristics for both groups must be presented.
  2. Add patient characteristics for the bulbar onset, limb onset, age, ALSFRS-R score, disease onset, NIPPV or TPPV status and etc. of ALS patients as a Table 1.
  3. If patients with ALS were using NIPPV, it is helpful to add a record of ventilator settings to understand the status of ALS patients who included in this study.
  4. In the introduction, it can be helpful to add a general explanation of ALS disease for the readers, using the reference below
  • The terminal latency of the phrenic nerve correlates with respiratory symptoms in amyotrophic lateral sclerosis. Park JS, Park D. Clin Neurophysiol. 2017 Sep;128(9):1625-1628. doi: 10.1016/j.clinph.2017.06.039. Epub 2017 Jul 1.
  • Amyotrophic lateral sclerosis. van Es MA, Hardiman O, Chio A, Al-Chalabi A, Pasterkamp RJ, Veldink JH, van den Berg LH. Lancet. 2017 Nov 4;390(10107):2084-2098. doi: 10.1016/S0140-6736(17)31287-4. Epub 2017 May 25.

- Terminal latency abnormality in amyotrophic lateral sclerosis without split hand syndrome. Park D, Park JS. Neurol Sci. 2017 May;38(5):775-781. doi: 10.1007/s10072-017-2842-8. Epub 2017 Feb 10. PMID: 28188450

  1. Why is FVC important in ALS patients, why a reduction in FVC occurs, and a description of the respiratory pathophysiology should also be added, using the reference below.

- Different characteristics of ventilator application between tracheostomy- and noninvasive positive pressure ventilation patients with amyotrophic lateral sclerosis. Park D, Lee GJ, Kim HY, Ryu JS. Medicine (Baltimore). 2017 Mar;96(10):e6251. doi: 10.1097/MD.0000000000006251.

- Prognostic value of phrenic nerve conduction study in amyotrophic lateral sclerosis: Systematic review and meta-analysis. Silva CS, Rodrigues FB, Duarte GS, Costa J, de Carvalho M. Clin Neurophysiol. 2020 Jan;131(1):106-113. doi: 10.1016/j.clinph.2019.10.016. Epub 2019 Nov 11.

- Application of different ventilator modes in patients with amyotrophic lateral sclerosis according to certain clinical situations: A Case Report. Park D. Medicine (Baltimore). 2017 Aug;96(34):e7899. doi: 10.1097/MD.0000000000007899.

- Under-recognized primary spontaneous pneumothorax in ALS: a multicenter retrospective study. Park JS, Do YW, Park JM, Seok HY, Park D. Neurol Sci. 2019 Dec;40(12):2509-2514. doi: 10.1007/s10072-019-03989-y. Epub 2019 Jul 2.

Author Response

Author's Reply to the Review Report (Reviewer 1)

There may be a problem with the reliability of the FVC measurement. If the bulbar palsy is severe in ALS patients, the reliability of the FVC value is problematic. For this purpose, the patient characteristics for both groups must be presented.

Add patient characteristics for the bulbar onset, limb onset, age, ALSFRS-R score, disease onset, NIPPV or TPPV status and etc. of ALS patients as a Table 1.

Response 1: Patients with ALS with bulbar onset were excluded from this group. Line 98-100

If patients with ALS were using NIPPV, it is helpful to add a record of ventilator settings to understand the status of ALS patients who included in this study.

Response 2: The ventilator settings are variable according to each patient and with the parameters of the FVC performed at the time of care. In addition, the aim of the study was only to classify patients as dependent on NIPPV or not.

In the introduction, it can be helpful to add a general explanation of ALS disease for the readers, using the reference below

The terminal latency of the phrenic nerve correlates with respiratory symptoms in amyotrophic lateral sclerosis. Park JS, Park D. Clin Neurophysiol. 2017 Sep;128(9):1625-1628. doi: 10.1016/j.clinph.2017.06.039. Epub 2017 Jul 1.

Amyotrophic lateral sclerosis. van Es MA, Hardiman O, Chio A, Al-Chalabi A, Pasterkamp RJ, Veldink JH, van den Berg LH. Lancet. 2017 Nov 4;390(10107):2084-2098. doi: 10.1016/S0140-6736(17)31287-4. Epub 2017 May 25.

- Terminal latency abnormality in amyotrophic lateral sclerosis without split hand syndrome. Park D, Park JS. Neurol Sci. 2017 May;38(5):775-781. doi: 10.1007/s10072-017-2842-8. Epub 2017 Feb 10. PMID: 28188450

Response 3: We appreciate all the suggested references. We modified the introduction according to the suggestions. Line 40-53

Why is FVC important in ALS patients, why a reduction in FVC occurs, and a description of the respiratory pathophysiology should also be added, using the reference below.

- Different characteristics of ventilator application between tracheostomy- and noninvasive positive pressure ventilation patients with amyotrophic lateral sclerosis. Park D, Lee GJ, Kim HY, Ryu JS. Medicine (Baltimore). 2017 Mar;96(10):e6251. doi: 10.1097/MD.0000000000006251.

- Prognostic value of phrenic nerve conduction study in amyotrophic lateral sclerosis: Systematic review and meta-analysis. Silva CS, Rodrigues FB, Duarte GS, Costa J, de Carvalho M. Clin Neurophysiol. 2020 Jan;131(1):106-113. doi: 10.1016/j.clinph.2019.10.016. Epub 2019 Nov 11.

- Application of different ventilator modes in patients with amyotrophic lateral sclerosis according to certain clinical situations: A Case Report. Park D. Medicine (Baltimore). 2017 Aug;96(34):e7899. doi: 10.1097/MD.0000000000007899.

- Under-recognized primary spontaneous pneumothorax in ALS: a multicenter retrospective study. Park JS, Do YW, Park JM, Seok HY, Park D. Neurol Sci. 2019 Dec;40(12):2509-2514. doi: 10.1007/s10072-019-03989-y. Epub 2019 Jul 2.

Response 4: Respiratory function impairment is important, however this article should highlight the importance of HRV as a predictor of prognosis for ALS in order to detect early abnormalities, estimate the prognosis and define the treatment strategy. We appreciate all the suggested references. Line 45-50.

Reviewer 2 Report

In this paper, the complexity of heart rate variability in amyotrophic lateral sclerosis (ALS) patients with different pulmonary capacities was evaluated. The conclusion shows that the complexity of HRV in ALS subjects with different pulmonary capacities increased through chaotic global analysis especially CFP5 and three of five entropies. It is very interesting and well organized. However, it has some problems that need to be solved further.

  1. The innovation of the article has not been reflected in both theoretical and experimental design. The author needs to revise the abstract and introduction to further highlight the contribution and innovation of this paper.
  2. The structure of the article needs to be improved, there are too many points, lack of integrity, such as Section 2 and 4.
  3. All Figures need to be modified, there are format error. It is suggested that the authors use the color ones.
  4. There is only one sentence in the conclusion, which is very poor. At least three points need to be carefully summarized and supported by key numerical results.
  5. All references are not in standard format. The authors should check and revise them carefully.

In order to qualify for publication in Entropy, the paper must be improved according to the comments to the authors.

Author Response

  1. The innovation of the article has not been reflected in both theoretical and experimental design. The author needs to revise the abstract and introduction to further highlight the contribution and innovation of this paper.

Response 1: We changes to the introduction. Line 40-53.

2.The structure of the article needs to be improved, there are too many points, lack of integrity, such as Section 2 and 4.

Response 2: We corrected the integrity between sections 2 and 4.  

3. All Figures need to be modified, there are format error. It is suggested that the authors use the color ones.

Response 3: Revised

4. There is only one sentence in the conclusion, which is very poor. At least three points need to be carefully summarized and supported by key numerical results.

Response 4: Line 339-344

5. All references are not in standard format. The authors should check and revise them carefully.

Response 5: The references were revised and corrected

Round 2

Reviewer 1 Report

Thank you for your effort. 

Reviewer 2 Report

Thank the authors for their efforts. The authors have adequately addressed all my concerns in the review, and did a good job to revise and improve the paper. The paper now is suitable for publication in Entropy in its current form.